# ‘This Adds Another Perspective’: Qualitative Descriptive Study Evaluating Simulation-Based Training for Health Care Assistants, to Enhance the Quality of Care in Nursing Homes

**DOI:** 10.3390/ijerph18083995

**Published:** 2021-04-10

**Authors:** Catherine Aicken, Lisa Hodgson, Kay de Vries, Iain Wilkinson, Zena Aldridge, Kathleen Galvin

**Affiliations:** 1School of Health Sciences, University of Brighton, Falmer BN1 9PH, UK; l.hodgson@brighton.ac.uk; 2School of Nursing and Midwifery, De Montfort University, Leicester LE1 9BH, UK; kay.devries@dmu.ac.uk (K.d.V.); zena.aldridge@dementiauk.org (Z.A.); 3Surrey and Sussex Healthcare NHS Trust, Redhill RH2 5RH, UK; iainwilkinson1@nhs.net; 4Brighton and Sussex Medical School, Falmer BN1 9PH, UK; 5Dementia UK, London EC3N 1RE, UK

**Keywords:** long-term care, nursing homes, implementation science, quality improvement

## Abstract

Much of the UK’s ageing population lives in care homes, often with complex care needs including dementia. Optimal care requires strong clinical leadership, but opportunities for staff development in these settings are limited. Training using simulation can enable experiential learning in situ. In two nursing homes, Health Care Assistants (HCAs) received training in clinical communication skills (Situation-Background-Assessment-Recommendation Education through Technology and Simulation, SETS: group training with an actor simulating scenarios); and dementia (A Walk Through Dementia, AWTD: digital simulation, delivered one-to-one). In this qualitative descriptive study, we evaluated the potential of this training to enhance HCAs’ clinical leadership skills, through thematic analysis of 24 semi-structured interviews with HCAs (before/after training) and their managers and mentors. Themes were checked by both interviewers. HCAs benefitted from watching colleagues respond to SETS scenarios and reported greater confidence in communicating with registered healthcare professionals. Some found role-play participation challenging. AWTD sensitised HCAs to the experiences of residents with dementia, and those with limited dementia experience gained a fuller understanding of the disease’s effects. Staffing constraints affected participation in group training. Training using simulation is valuable in this setting, particularly when delivered flexibly. Further work is needed to explore its potential on a larger scale.

## 1. Introduction

### 1.1. Background: The Ageing Population and Care Home Sector in England

England has an ageing population, and a growing number of older people live in care homes (at 329,000, more than three times the number of hospital beds) [1,2]. Half of care home residents aged 65 years or older have complex health and social care needs [3] with the majority having multiple co-morbid conditions [4]. Increases are expected in the number of residents with complex care needs, the number of years of old age spent in dependency, and the size of the care sector as a whole. Across the UK it is estimated that 311,730 care home residents have dementia, of which 131,230 live in nursing homes (where there is 24-h nursing provision on site) [5,6]. People living with dementia have, on average, more than four chronic conditions [7].

Nursing homes in the UK are situated within social care, or private care, and outside of the country’s National Health Service (NHS). (This is for historical reasons: in 1948 when the NHS and Social Care were established as two separate public services, life expectancy in general, and for disabled adults requiring care, was shorter, and dependent adults were more likely to be cared for solely by their families. As this situation has changed, demand for social care has risen hugely, but whilst NHS care is free at the point of use, means-tests are in place for social care [8]). Yet nurses and healthcare assistants (HCAs) in nursing homes have to manage residents’ very complex care needs, including dementia, comorbidities and frailty. There is a lack of consensus on how best to meet these needs, and support for care homes from primary and secondary healthcare services is variable, often leaving care homes isolated from the wider healthcare system [9]. Delayed provision of healthcare and support leads to an increased risk of unplanned hospital admissions, morbidity and mortality [10,11,12,13], and there is often poor acknowledgement in care homes of adverse events (e.g., a fall or infection) that can be indicative of decline in frail residents [14]. Over half of older care home residents lack appropriate access to the NHS services they need, and consequently many are inappropriately admitted to hospital [15]. Compared with people of the same age living in the community, older care home residents are 40–50% more likely to attend Emergency Departments or be admitted to an acute hospital bed, and are less likely to have planned hospital admissions or attend out-patient appointments [13]. The latter necessitate liaising with NHS medical and nursing staff, at the individual and institutional level. When care homes and health services work closely together, impressive results have been demonstrated, e.g., reductions in urgent admissions to hospital of 30% or more [16].

### 1.2. Skills and Staff Development Needs of Health Care Assistants in the Care Home Sector

Strong clinical leadership—being able to recognise changes in residents’ health status and having high-level decision-making skills regarding appropriate care needs—is necessary for delivering high quality care. The range of clinical leadership skills care home staff require includes complex communication skills to communicate with older people with a diverse range of sensory needs; end of life care skills; specialist dementia care; and knowledge and skills in assessment related to many conditions and comorbidities with a very complex client group [17]. Skills gaps within the sector are linked to problems in education and training, and to challenges in staff turnover and recruitment [17]. Staff retention in the UK’s care home sector is poorly understood, but recognised to be problematic, with high rates of vacancies [15,18,19]. Problems with clinical leadership in the sector can lead to lost productivity, high replacement costs (including training), low staff morale, low job satisfaction, and inconsistent or compromised quality of care [15,17,20].

Health Care Assistants (HCAs) are the main providers of direct care to nursing home residents [21], and in the UK, no degree or professional qualification is needed to work in this role (although many care home providers now require non-professionally registered staff to complete a care certificate [22], and may ask for relevant care experience). The majority of care staff in the sector’s workforce are low paid, low status and have no clear career path. Within social care in the UK, there are low levels of literacy and numeracy; furthermore, many staff have English as a second or additional language [23], as there is a significant reliance on migrant workers [24]. In the absence of mandatory entry qualification requirements, and with disparities in basic skills, new starters often lack appropriate leadership skills and subsequently learn on the job [15]. Despite the need for skills training, opportunities for staff development in the care home sector are often sparse; there is a shortage of funding to provide training, particularly non-statutory, advanced or specialist training [15]. Educational opportunities need to be more clinically relevant and tailored to the care home setting [17,25,26,27,28,29]. Person-centred care—i.e., care that meets individuals’ needs and preferences, which in practice involves relationship-building [30]—is widely recognised as desirable yet may be overlooked in task-oriented work [31]. Education and ongoing staff training that fosters person-centred care can facilitate the development of a culture of person-centred care within healthcare settings [32,33]. Being supported to provide person-centred care may benefit care staff (as well as residents), through greater satisfaction with their work [20,30].

SBAR (Situation, Background, Assessment, Recommendation or Request for action) is a widely used situational briefing model which provides a concise, predictable structure to communication about patients’/residents’ health situations between people involved in their care [34,35,36]. (For examples of SBAR use in practice, see: [37,38]). A systematic review of SBAR’s impact on patient safety found moderate evidence for an improvement, especially when used to structure communication over the phone [38]. This review included only three studies in care home settings, each with a very specific focus (reducing hospital transfers of nursing home residents [39]; a warfarin communication protocol [40], and transfers, hospitalisations, and 30-day readmissions from long-term care to acute-care [41]). Despite its wide use in clinical settings there is limited research in care home contexts, and high-quality research on SBAR is lacking (only one controlled trial [40] included in the systematic review was ‘strong’ in quality) [38]. SBAR can be taught in diverse ways, for instance through an online module, lecture, written material or simulation.

### 1.3. Effective Learning in the Care Sector, and the Potential Role of Simulation

In traditional views of workplace learning, development of practical competencies involves learning and gaining experience in order to obtaining attributes (appropriate attitudes, conceptual knowledge, and practical skills) [42]. Dall’alba and Sandberg, however, emphasise the importance of developing skills in context, and of embodied understandings of practice as ways in which learners develop [42]. Specific forms of learning may be preferred by learners working in settings where they have a considerable need for interaction and construction as their expertise grows [43], such as care homes.

A systematic review [44] that aimed to identify characteristics of effective dementia education and training for health and social care staff across service settings, found that the training/education most likely to be effective included several important features. It needed to be relevant and realistic, tailored to the roles, experience, and practice of learners. It should include active participation and underpin practice-based learning with theoretical or knowledge-based content. It was also effective when experiential and simulation-based learning included adequate time for debriefing and discussion, and was delivered by an experienced trainer/facilitator who was able to adapt it to the needs of each group. Effectiveness was also attributed to not relying on written materials or in-service learning as the sole teaching method. Learning activities that supported the application of training into practice, and provided staff with a structured tool, method or practice guideline to underpin care practice, were also shown to be effective.

Simulation-based education can have many of these characteristics, and is effective for practice-relevant training of the health workforce [45]. It is increasingly popular in nursing education, enabling students to practice their clinical and decision-making skills through real-life situational experiences [46,47]. Virtual patients expose learners to simulated clinical experiences, providing mechanisms for rehearsing information gathering and clinical decision making in a safe zone [48]. Whilst there is a growing evidence base for simulation-based education with healthcare professionals and in acute settings (including large-scale evaluations [49]) there is less evidence from care settings or with non-registered care staff such as HCAs. Our study addresses this gap.

### 1.4. Study Aims

We aimed to investigate how simulation-based training can enhance the clinical leadership skills of HCAs within nursing homes, in order for them to improve the quality of life of people in care. Our study explored the need for and potential role of simulation-based training, and qualitatively evaluated two types of simulation-based training, exploring their acceptability to HCAs, and the impact that they may have on HCAs’ practice (as reported by HCAs and their colleagues):
‘A Walk Through Dementia’ (AWTD) interactive smartphone app, which uses virtual reality (VR) to simulate the experience of having dementia [50], implemented on a one-to-one basis. AWTD is self-contained and does not require input from a trainer.SBAR Education through Technology and Simulation course (SETS). The SETS course was delivered to a group of HCAs by a consultant geriatrician who is an experienced medical educator (and a Fellow of Advance HE, the UK’s Higher Education Academy). SETS uses an actor to simulate scenarios appropriate to the settings’ needs. The training focused on deterioration in health.


The two types of training were chosen because they are both relevant to HCAs working in nursing homes for older people, yet they are very different, with contrasting ease of implementation and use of resources. Our study therefore offers an opportunity to generate tentative findings about the role and value of simulation-based training per se, and the role and value of each, including preliminary evidence of feasibility of implementation, with HCAs in nursing home settings.

## 2. Materials and Methods

Our two-phase evaluation study used a qualitative description approach [51] to explore the impacts of training on HCAs. This method was suited to our study’s aim, as it enabled us to generate a description of the role and value of simulation-based training and its impacts on HCAs’ work, from the perspectives of those working in nursing homes. Compared to other qualitative methods, qualitative description is less ‘theoretical’ [51], which suited our study as we sought to stay close to the data, imposing minimal interpretation on it.

### 2.1. Study Population, Setting and Recruitment

The study took place in two nursing homes for older people, run by an independent care organisation which operates multiple care homes in southern England. The homes, both located in villages, have 48 beds and 60 beds, and both have a dedicated wing for residents with dementia although not all residents have this condition. The offer of free staff training constituted an incentive for managers, staff, and the organisation as a whole, to engage with the study.

Care home managers introduced us to HCAs (we use this term inclusive of care assistants and senior care assistants) who they considered would benefit from training, and to staff in supervisory and/or mentorship roles (including Clinical Lead Nurses and Assistant Managers) whom we refer to as ‘mentors’ for brevity. Prior to commencing the study, we understood from our initial contact with the care home organisation that mentors were members of care home staff whose primary role is to support the development of front-line care staff. However, when we visited the homes, we found that the term ‘mentor’ was not used by staff, and there were no staff in this dedicated role. Assistant Managers, Clinical Lead nurses and some HCA supervisors identified HCAs’ training needs and supported their development as part of their work in the care homes.

We requested that staff were released from their duties for the duration of recruitment discussions, interviews and training, i.e., they should not forfeit their breaks due to study participation. Potential participants were offered a Participant Information Sheet to read and keep, and given the opportunity to discuss the study with the researcher and ask questions. Participants signed an informed consent form prior to participation in a voluntary, confidential interview.

### 2.2. Data Collection

One-to-one semi-structured interviews were conducted in private rooms in the homes (an empty lounge, staff room or office), and audio-recorded with consent. Researchers (CA and LH, both trained and experienced in qualitative interviewing and analysis) additionally reassured staff of our independence from the care home organisation, and that no individually identifying information would be shared with employers or published.

Figure 1 outlines the study design, which included interviews with HCAs before and several weeks after receiving training, and interviews with mentors and managers over the duration of the study. We sampled purposively by job role, and in Phase 2 sought only to interview those who had participated in AWTD and/or SETS training. Data collection materials are provided in Appendix A. Training was prioritised for HCAs, but we allowed other staff to participate where desired and feasible.

#### 2.2.1. Steps Taken to Enable Participation of HCAs

Mindful of high staff turnover within the sector, high use of agency staff and changeable shift patterns, we knew at the outset that it would be challenging to retain the same HCAs in Phase 1, training, and Phase 2 of the study. This influenced our study design in three ways. We made a priori decisions (i) to seek brief quantitative and qualitative feedback from HCAs immediately after each training session via an anonymous Feedback Form (see Section 2.2.3, and Appendix A) in case we could not achieve Phase 2 interviews; and (ii) to conduct Phase 2 interviews with HCAs who had participated in training whether or not we had not interviewed them in Phase 1. In Phase 2, a researcher provided a list of HCAs who had participated in training to the managers and sought to visit the homes when these people were working. Despite multiple visits, these people were often unavailable due to rota changes, sickness, annual leave, and being too busy to be interviewed. We took an additional step, (iii) relaxing the requirement of Phase 2 interviewees to be HCAs: we interviewed any staff member who had participated in training, asking those in other roles to reflect on the impact of the training on their HCA colleagues.

#### 2.2.2. Manager and Mentor Interviews

Interviews with managers and mentors covered the following topics: the manager/mentor’s role in supporting HCAs; perceptions of HCA training needs; how HCAs could best be supported; barriers and facilitators to HCA training and development (Appendix A, Appendix A). Manager and mentor interviews were not restricted to Phase 1 or Phase 2 but could occur at any point in the study, to minimise the impact of data collection on busy staff.

#### 2.2.3. Pre- and Post-Training Interviews and Feedback Questionnaire

Topic guides from Phase 1, and for mentor and manager interviews were developed based on an understanding gained from the literature about care staff’s training needs, and the sector’s challenges with training and retention (see Introduction). Interviews with HCAs prior to training covered: experience of, and route to, care work; views/feelings about care work; self-identified training/development/support needs; experience of, and attitude to, training; experience of working with residents with dementia; experience of communicating with registered healthcare professionals about residents’ health (Appendix A). Where senior HCAs mentioned that they had supervisory/supportive roles over other HCAs, we additionally asked relevant questions from the mentor interview topic guide.

Immediately after each training session, HCAs were asked to complete a questionnaire which invited brief quantitative and qualitative feedback about the training (Appendix A).

Staff who had participated in at least one training session were invited to another one-to-one interview (Appendix A). A 4–6 week gap between training and the interview was planned to allow sufficient time to have elapsed for HCAs to put the training into practice, but for memory of the training to be still fresh. Phase 2 topic guides were designed to explore staff’s experiences of training, and any impacts of this training. This interview covered: how things have been recently at work; views/feelings about care work; reflections on the training received; and (depending on which training had been undertaken) recent experience of: working with people with dementia (after AWTD training) and/or communicating with registered healthcare professionals about residents (after SETS training). Staff who were not HCAs were asked to reflect on the impact of the training on their HCA colleagues.

### 2.3. Training Implementation

AWTD was implemented in a quiet room in the home, using the research team’s smartphone, placed within a cardboard headset (or held by the user, if preferred, or if the headset pressed on the phone’s off-button). Researchers invited HCAs to attend one by one during their shift, and managers and senior staff encouraged participation. A researcher remained present during use of AWTD, in case of problems with using the app or headset, and for safety (as the headset covers the user’s eyes). Three interactive scenarios were available, simulating experiences of dementia (a street setting, a shop, making refreshments for visitors at home). Researchers encouraged HCAs to try at least two, which took less than half-an-hour, although we allowed more time if desired. We provided participants with the website address from which AWTD can be viewed online or downloaded as a free app.

SETS training required participants to attend a half-day group session. Researchers liaised with home managers to identify dates when up to 10 HCAs could participate, and one SETS session was delivered in each home. SETS required use of a training room (we used an empty lounge), and a vacant bedroom where scenarios took place. The trainer (IW) began by describing SBAR and its utility as a communication tool. Four scenarios focused on deterioration in health in a care home resident living with dementia or a cognitive impairment, played by a professional actor (experienced in acting SETS role-plays). In each role-play, two training participants entered the ‘resident’s’ room, whereupon the scenario began, and other participants observed the scenario via video-link. In each scenario, participants were told that they should communicate with at least one other person, using SBAR to structure this communication. The participants chose who they would communicate with (e.g., the resident’s GP, social worker, or relative), and the trainer acted as this person. This communication should be about the ‘resident’s’ state of health and should include a recommendation or request for action (e.g., that the GP needs to check on the resident on their weekly visit to the home; that they would like the relative to give the resident a reassuring phone call today). After each scenario the trainer facilitated a discussion which followed the ‘TeamGAINS’ structured debriefing model. This debriefing model was chosen as it has been designed specifically for simulation-based training and is suited to situations where there is an ‘expert model’ (in this case, SBAR) to refer back to [52]. During the debrief, sections of the recording of the simulation video were played when this added educational value.

### 2.4. Data Management and Analysis

Interview recordings were transcribed by a commercial transcription agency, and checked for accuracy by researchers. Thematic Analysis [53] was chosen as it is a theoretically flexible, transparent method, suited to the analysis of qualitative data in evaluation research. The two researchers who conducted the interviews read interview transcripts repeatedly for data familiarisation. One researcher led the analysis (using tables in MS Word and Excel for data management), and the other checked the themes. Identification of the main themes was driven largely by the research questions, whilst sub-themes were identified inductively, emergent from researchers’ interaction with the data. After identifying candidate sub-themes, we searched for negative cases, in order to refine the sub-themes and give greater depth to the analysis. Interview data formed the bulk of the qualitative data, supplemented by free-text responses from Feedback Forms. Visual inspection of the quantitative data from the Feedback Forms indicated that it corroborated (or did not contradict) the findings of our qualitative analysis. (Due to the small size, exploratory nature of the study, and successful implementation of Phase 2 interviews, quantitative analysis of the data in the Feedback Forms was not undertaken).

## 3. Results

The primary, descriptive themes were: (1) experience of working as a HCA; (2) training provision, training needs and responsibilities for staff development; (3) experiences of taking part in simulation-based training; (4) impacts of the simulation-based training. Appropriate to this study’s focus, the first two themes are outlined briefly for context. Themes 3 and 4, which speak to the study’s aim, are described in depth, with sub-themes, the names of which clarify whether they apply to AWTD, SETS, or both.

In Phase 1, 15 interviews were undertaken (Table 1): 10 with HCAs (of which four had some role in supervising other HCAs); two with assistant managers of which one was also clinical lead; one clinical lead who was not in a managerial role; and two managers.

Researchers showed the AWTD app to 14 HCAs, and SETS participants included 12 HCAs (some completed both trainings whilst others did only one). It was considerably easier to implement AWTD than SETS, due to its brevity, one-to-one nature, and minimal resource and planning requirements. AWTD participants often watched more than one scenario, although rarely all three, citing the need to return to work. We obtained 25 completed Feedback Forms from HCAs who participated in training (one HCA returned to her work straight after AWTD and did not return her form despite reminders). Researchers and managers had hoped that more HCAs would participate in SETS (neither course was full), although at least two HCAs came to work on their days off to participate.

In Phase 2, we undertook nine interviews: four with staff who had experienced AWTD (two HCAs, one nurse and one activities co-ordinator, the latter two with experience of working as HCAs), and five with staff who had participated in SETS (quotes from Phase 2 interviews are labelled ‘postSETS’ or ‘postAWTD’, indicating which training participants had done.) Whilst all HCAs who we interviewed had some awareness and training on dementia, no HCAs had come across SBAR before. In Phase 2 interviews it was often difficult to distinguish between their views on SBAR as a communication tool, and views on the use of simulation in SBAR training.

### 3.1. The Experience of Working as a Health Care Assistant

HCAs described how their duties sometimes gave them little time to sit with residents, or to take their entitled breaks. The HCA role was described as emotionally and physically tiring, with risk of burn-out and high staff turnover. In the context of these challenges their dedication to a caring role was apparent; HCAs often explained how it was not possible to do the job if you did not enjoy care work: ‘*I like to take care for the people. If you don’t like, you can’t have this job’* (CA08).

HCAs described diverse career trajectories, with and without ambitions to become registered healthcare professionals. Several described being qualified nurses in their countries of origin, and working as HCAs whilst gaining the appropriate English language and other accreditations to become registered nurses in the UK.

Within the nursing homes, HCAs described clearly defined roles in terms of who communicated with GPs, ambulance staff or other professionals. Nurses and managers did this; HCAs rarely communicated with external health professionals about accidents/incidents or a resident’s clinical need. Communication and working relationships between different staff roles within the homes were described positively.

### 3.2. Training Provision, Training Needs and Responsibilities for Staff Development

#### 3.2.1. Provision of Training and Development Activities

Care home managers described how HCAs were required to participate in a 4-day induction (encompassing mandatory training) prior to working in the homes, and regular ongoing training. The company made further training (including updates to mandatory training) and career progression opportunities available to staff through training delivered in person at the company’s Academy and sometimes in the homes; and through online training, which now encompassed a large proportion of the available training. They described how staff and managers could request further relevant training, and individuals might then be sent to other training providers or Academy trainers might visit the homes.

Managers and clinical leads (nurses) described how they identified training needs through supervision of HCAs, referring to this as ‘*supporting*’ HCAs to fulfil their roles.

#### 3.2.2. Modes of Delivery of Existing Training

Online training could be accessed via staff’s own devices, or via company computers in the Academy or workplace. As such, it could be completed in paid work time, or outside of work time unpaid: ‘*obviously we can’t pay them, because we don’t know how long* [it took]’ (CA02). In contrast, attending face-to-face training would be paid whether or not it took place during staff’s work time.

#### 3.2.3. Attitudes to Training and Development; Barriers and Facilitators

A range of attitudes towards training and career progression was described. Whilst one Deputy Manager and former HCA described ‘*if it’s free and they provide it you might as well do it*’ (CA01), a view accepted as ideal by many HCAs, she described that some HCAs would do little or no non-mandatory training. Managers and most staff interviewed described how the company encouraged staff development and progression, including through National Vocational Qualifications (NVQs). One manager explained how employment contracts now stated that staff had to pay back training costs if they left the company within a few months of gaining their NVQ, adding that this change had reduced staff retention problems.

Managers tended to discuss how accessible the training was, and how they and the company sought to enable this. On further discussion, they identified some barriers to training uptake: older staff could feel ‘*daunted*’ by new training requirements (stemming from legal requirements affecting the sector), or ‘*frightened*’ or ‘*worried*’ by the idea of completing training online; staff with children, and those who wished to take longer to read and absorb new material might prefer the flexibility and relaxed pace of doing their online training from home.

There was a slight contrast with the views expressed by HCAs, none of whom expressed that the training was daunting. Instead, they discussed that although computers were available in the homes, it was challenging to find time to use them during a shift, which meant that their training was frequently unpaid. Accessing the Academy was difficult for HCAs who did not drive, as ‘*there’s no public bus*’ and although the organisation provided transport ‘*the driver leaves erm at three and sometimes the training finishes at four. So it’s a problem getting back*’ (CA12), and barriers included needing childcare.

### 3.3. Experiences of Taking Part in the Simulation-Based Training

#### 3.3.1. Disorientation and Fear as Ambivalent or Positive Features of AWTD

Some users felt disorientated by the AWTD app (especially when using the headset), a feeling they described as unpleasant and unanticipated. The most extreme example of this was nausea:

*it made me feel quite, erm … disorientated. […] … I thought I was falling off the chair when I went to look round. I actually felt I was falling over on that* [hole: simulated experience of perceiving a puddle as a hole] *[…] I was surprised actually. I was not expecting that when I started the video. Well I didn’t know what to expect before I started. I didn’t think it would have such a physical reaction from me like feeling sick*. (LH02 postAWTD)

Whilst disorientation was experienced negatively, it was simultaneously understood as a positive feature of the training: ‘*it was really, really good because it was so disorientating and quite scary*’ (LH05 postAWTD). Disorientation provoked sympathy about what life might be like for a person with dementia. Despite prior knowledge and experience of working with people with dementia, AWTD was ‘*quite an eye opener*’:

*a big reminder for me or acknowledgement for me, that it’s actually physical feeling as well […] I had to like re-orientate myself. And if you have got dementia you can’t necessarily do that*. (LH02 postAWTD)

*you work in dementia, and I have done for four year, but you know things, you know that it can impact them in the way think, you know levels of confusion exists with the most simplest of things, but you gave us those goggles* [headset], *you still know all that, but you see it differently because you see it through your eyes. It is like it is giving you a taste of what it is like, what they see and perceive things. Very strange, good, but scary feeling...*
(LH05 postAWTD)

One interviewee explained how the person with dementia’s internal dialogue (which could be heard whilst using the app) enhanced the visual simulated experience:


*So it was bringing it to life, erm, the hesitation in their voice, what was confusing, how scared they are feeling. That’s why it is good to train with it I think.*
(LH06 postAWTD)

#### 3.3.2. Learning from Watching Others, and Discomfort about Being Watched (SETS and AWTD)

Nervousness and discomfort were dominant feelings described by SETS participants, including those who described the training positively:


*It was interesting and I’m glad I got to do it even if I was a little nervous to start with. Once I got over that I really enjoyed taking part.*
(LH08 postSETS)

In particular, staff expressed nervousness about taking part in SETS role-plays with their colleagues watching. Watching others go first could aid learning, or could prolong this nervousness such that it interfered with the learning process, as these contrasting quotes illustrate:

*…I had time to think and learn even from the previous scenarios. Which helped me a lot. I would have been too nervous to go first* [laughs]. (LH03 postSETS)


*the first one I was nervous watching and I’m not sure how much of what they were doing I took in because I kept thinking of my turn next […] after I had my go I was much better, I had relaxed and seen what it was like so erm I think I could relax and not worry and I could erm…erm…take part a lot more than I did in the first feedback before my go.*
(LH08 postSETS)

Once their turn at the role-play began, the realism of the scenarios could help participants relax and become less aware that their colleagues were watching:


*The actor was really good. He played the situation realistically erm so I think you just went into a realist reaction. I mean I was aware that I was being watched but as it played out erm…I think I must have forgotten about the cameras, well, until I got back in the room and could see the screen.*
(LH08 postSETS)

The trainer’s manner helped lessen participants’ sometimes considerable apprehension about receiving feedback on their role-play:


*although I was intimidated by it, he was very good and fun with it, putting nerves at ease.*
(LH07 postSETS)

*I think the tone of it was just set by the way he approached it. Steady, easy feedback, nothing horrible you see. You weren’t told you were wrong, we were just shown other ways to do it, which may not have been thought of… […] …and he told a little bit about the scenarios, so* [debriefs] *weren’t necessary all about what we were doing* [in the role-play](LH08 postSETS)

Participants generally considered that SETS was a valuable learning experience despite the discomfort of being watched, with the feedback after each role-play providing space for reflection and further learning:


*I was uncomfortable doing the role-play but liked watching and talking through our actions at the end. That was really helpful, we learnt about the conditions more or the cause of situations more, the theory if you like. The trainers shared nuggets of information why something might cause an illness or a reaction. That was really interesting.*
(LH07 postSETS)

*…it was useful, so although it is uncomfortable, on reflection, once you had a go, you relax and talk and you start seeing the pieces come together about what you did and did not do*. (LH03 postSETS)

One HCA, who also described the experience as ‘*uncomfortable*’, commented how she would have preferred to know in advance that she would be taking part in an observed role-play: ‘*I didn’t like that it was being watched. I think I would have liked to have known that before we got started, not when I was in there*’ (LH09 postSETS)

Discomfort about being watched was unexpectedly experienced by some AWTD participants, but without the benefit and reciprocity experienced by SETS participants. While researchers were showing AWTD, other staff occasionally entered the room to obtain equipment or to check whether the previous participant had finished. HCAs who were using the headset found it unsettling as they could not see who was there, and ‘*they distract you as well and they laugh at you*’ (LH01 postAWTD). In contrast, SETS role-plays were uninterrupted, perhaps because it was clearer to colleagues that a training event was taking place.

#### 3.3.3. Realism and Learning through Practice (SETS)

Use of scenarios to teach SBAR was helpful to people who preferred to learn through doing:


*I’m quite a practical person rather than being theory based. So that helped me to learn.*
(LH08 postSETS)

Later in her interview the same HCA expanded on this, explaining how the trainer:


*gave us some background about why a resident might act or be acting a certain way, which really was interesting to me […] and because it related to a practical situation which we had just acted out so it stays in my head better. Erm, probably stuff I never had thought of before, but actually was factual and interesting. Not boring at all because it could be practically applied to what we had gone through.*


She explained that although ‘*you can learn a lot from reading*’, if it is *’too hard or too much jargon you don’t always understand it and it becomes impossible to see, erm, you can’t then link it to things. The way the trainer did it was to apply that knowledge and for me that worked so well. Helped me learn it anyway and I don’t learn easily.*’ (LH08 postSETS)

Whilst a colleague explained that the simplicity of the SBAR acronym made it easy to remember, she added that ‘*because we had the practical exercise and then it just stayed with us*’ and ‘*then we discussed it, I think it was better that way*’ in comparison to reading about SBAR:


*otherwise it is just a lot of words, not so helpful. Helpful but not in the same way. We also had the theoretical part first and then that helped because we need to see how, and then the case studies, everything was really good. It all come together.*
(LH03 postSETS)

Interviewees agreed that the SETS scenarios effectively evoked real situations that might occur in the homes, although HCAs expressed differing views about how likely they were to be the ones to take the lead on communicating in these situations (discussed below). During the training and in post-training interviews, participants remarked on how effectively the actor mimicked an unwell resident. When he responded to their questions ‘*he had you know the physical response as well so it was really, it felt like a real situation*’ (LH03 postSETS).

The experience of SETS realistic scenarios was concisely described by this HCA:


*When you do a scenario, you act out what you would do. What is natural for you to do. Then you’re back into the room where the trainer is and everyone else too and you talk about it then. Really eye opening actually.*
(LH08 postSETS)

#### 3.3.4. Applicability of SETS and AWTD Scenarios to HCAs

Although none of the AWTD scenarios were set in a care home, no AWTD participants questioned the applicability of the app or its subject matter—dementia—to HCAs’ work. In contrast, divergent views were expressed about the applicability of the SETS scenarios, and SBAR, to HCAs. According to one HCA (a nurse in her country of origin), the scenarios were ‘*real situations that we are put through every, every day in our jobs so it helps us see what to do*’ and SBAR was ‘*just a communication tool so we can use it everywhere*’ (LH03 postSETS). However, another HCA remarked that she only appreciated the possible applications of SBAR after the training, when she discussed it with colleagues. For her, and some other interviewees, the sticking point seemed to be that HCAs in the homes do not usually relay information to outside health professionals This limited the apparent relevance of some scenarios:


*What you do on a daily basis, it didn’t come up, I’m not sure that they do scenarios around that but we didn’t experience it. I have struggled to use it every day since, in my job, so I think that would have been really good for me to have seen, a different practical side to the tool not just in an incident*
(LH07 postSETS)

Others discussed the applicability of SETS to situations that they deal with (discussed further in Section 3.4).

#### 3.3.5. Usefulness of Further Resources (AWTD and SETS)

No additional resources were provided with AWTD. When asked whether further resources would be helpful there was a general consensus that AWTD was ‘enough’ (as a complement to the dementia training that all HCAs had already received), but possibly with more time available to explore the app fully. In post-training interviews, none of the staff reported having looked at AWTD again, although some had recommended it to others.

Printed and online resources were available to complement SETS, which HCAs appreciated. In post-training interviews, some participants described having used these, and some had not.

### 3.4. Impacts of the Simulation-Based Training

#### 3.4.1. Insight into How a Person with Dementia May Experience the World (AWTD)

HCAs described how the AWTD app gave them a new perspective on the experience of living with dementia—even when they already knew about dementia and its effects, had worked with people with the condition, and considered themselves caring and compassionate:

*…you hear about what dementia is about […] But it is always harder to appreciate what that actually might be like from a different point of view. Seeing it is so different, it’s weird, yeah, so strange to feel like you’re in that situation. That is different from hearing something and it’s not really impacting you, or your body. I mean, we hear information and obviously you can empathise, understand and digest, but you never truly appreciate. The app, those videos gave you an impression, a taste of what it could be like*. (LH01 postAWTD, former HCA)


*it is difficult to get into a mind of someone with dementia, you hear about it, you understand the mechanics of the disease, but experience and the way the app makes you feel, it gives you those sense of disorientation, confusion, even the way you see things it changes that. I mean you can’t feel that when you read something about dementia. You understand its impact, but feeling it I think had a different kind of impact. I think it could even teach you more about the way you care, if you reflect, it could make you stop in your tracks, examine what you do and that is for all the work you do with all the residents I think, not just those suffering with dementia*
(LH02 postAWTD, Manager)

AWTD gave participants a sense of the embodied nature of living with dementia, as they felt a physical reaction to seeing things that disorientated or frightened them:

*it’s physical as well as just like you know a perceived thing. You body reacts with this condition, your brain reacts*. (LH02 postAWTD)

They described greater insight and understanding of the possible experiences of people with dementia, as a result of AWTD: ‘*I was a bit more compassionate to people I think afterwards’,* having realised ‘*how terrifying it must be for them.*’ (LH05 postAWTD)

A more experienced staff member, currently a nurse in the home, described that her awareness of dementia and its effects was quite high, but that she thought the app would be helpful to less experienced people. The training she had received previously was detailed:


*…but not as erm, how do you call it, not as helpful as this because it really shows you what’s happening to them I think, like real life what happens to a person who has dementia.*
(LH06 postAWTD)

#### 3.4.2. ‘*It Puts You in Their Shoes*’—Enhancing Person-Centred Dementia Care (AWTD)

By placing the carer in the position of a person with dementia, staff described how AWTD reminded them to be more patient, taking into account the different reality that some residents may experience:


*you’re carrying out a job, you have tasks to do to meet individual needs, and you want to do the best to meet those needs for every resident, you don’t not want to fulfil the simple things, but it, erm, it does mean sometimes, sometimes forget what the experience is. It takes things like this to remind you. […] …it gives you some appreciation of what they live with all day and every day. We get to go home and be ourselves, and forget, but that doesn’t happen for some of the residents does it.*
(LH02 postAWTD, manager)

*…since then I think I’ve viewed things differently and I think I treat PWD* [people with dementia] *as differently now because you get a bit more of an understanding they are not being difficult or trying to be annoying, you know they don’t know. It’s just they don’t know, the surroundings for them are completely different to what we can see.*
(LH05 postAWTD)

#### 3.4.3. AWTD’s Subtle Impacts on Practice

Staff described a qualitative change in their approach to people with dementia after experiencing AWTD, although they were often quick to explain that they were already performing their roles well:

*…we all know what to do and what is at the centre of our work. This just adds a layer to it*. (LH02 postAWTD, manager)

Responding to whether what she did has changed, one HCA explained:

*Well yeah and no really. What I do practically hasn’t changed. The needs of the patient has not changed, but I think my perception of the disease has. Like I said about* [own family member with dementia]. *I think it has now made me stop and think more when there a situation and say to myself, it’s not them, it is the condition. That is what I really think has changed.*
(LH05 postAWTD)

This very slight defensiveness was echoed in her colleague’s account, where she acknowledged the frustrations carers may feel working with people with dementia:


*I don’t think [I am] any more confident, just appreciate, awareness to see it differently, makes you more patient, even though I am a quite a patient person anyway. But I think it’s just given me more patience […] it does obviously get frustrating for everyone, I think people don’t always admit it. But it does, it gets like but I think once seeing it through their eyes, it is … it does extend that patience a lot.*
(LH01 postAWTD)

HCAs offered few practical examples of how their practice changed, but described reflecting on current and past actions. For instance, one described how she used to take a lady with Alzheimer’s disease, who was unable to communicate verbally, on regular trips to a café. After using AWTD, she reflected that although ‘*me and her family thought it was a good outing, she might have inside been terrified*’ (LH05 postAWTD). Another described how staff sometimes ‘*struggle*’ getting residents into the lift. She now realised that the gap at the entrance to the lift might be perceived as:


*a massive, like a hump in road kind of thing. So that’s why they were like hesitating trying to step over it. […] So it did open my eyes in that sense, erm… yeah they see what we don’t kind of thing. It is like you can know that, but when you experience it, like the app, it adds a level of understanding that you did not before.*
(LH01 postAWTD)

She went on to explain how she intended to use this understanding, by putting a picture in the back of the lift to for residents to look at, ‘*so it kind of like stops them getting stressed*’ when entering the lift. She was confident that the manager would allow this when they had time to implement it.


*I can see the relevance of using it for anyone having to deal with a relative or have to work with dementia patients. It is so easy to forget in the moment what they might be experiencing, or not have full understanding of what they are going through. This, this erm, stops you in your tracks and maybe examine how you approach things.*
(LH02 postAWTD)

#### 3.4.4. SETS and Efficient, Organised Communication

Some interviewees found that learning to use SBAR, through SETS, had helped them to communicate in a way which was ‘*more methodical*’ and organised:


*Just to have you know organised approach to every situation, just to think well what do I have here? What do I know? How am I going to report it and afterwards how am I going to resolve this so just to have an organised thought. I think that’s better if we want to give information and don’t forget anything.*
(LH03 postSETS)

Some interviewees were more sceptical about the training’s impact on their own practice, but despite this, observed changes in colleagues:


*…before she would be so rushing information, she’s much better now at giving it […] You [interviewer] are going to meet people and they’ll say it has done nothing for them but I can tell you from watching, working with them that yes it certainly has.*
(LH04 postSETS)

For one HCA though, the scenarios did not help her learn much ‘*I think I get what I’m doing, I’m good at my job and I know how to do it. It might have been [useful] for others though.*’ She described how she had been doing her job for a long time ‘*and this just seemed to be telling me what I know*’ (LH09 postSETS).

Another HCA explained how the video of her role-play showed her standing over the ‘resident’:


*that showed me how I was doing something I actually didn’t think I did do. Then you got suggestions about how you could do it differently. That made me really think about situations ever since. Not in incidents only, but just daily. I asked myself a little while after that, maybe the next day, was I standing rather than bending or erm sitting down, was I looking them in the face and eyes. So I think I corrected myself, made sure I took on board what was told to me. It was also not because I felt I was doing it badly, it was just that I thought it was better to do it another way. I learnt something that helped me communicate better, or maybe put the resident at ease.*
(LH08 postSETS)

As in the previous sub-theme, this HCA was keen to clarify that her current practice was adequate.

#### 3.4.5. SBAR’s Fit with Roles and Processes within the Homes

Whilst some identified that they might use SBAR when communicating with their nurse colleagues, for others, the fixed roles regarding communication in the homes (see Theme 1, Section 3.1) made some SETS scenarios seem inapplicable:


*if there could be some examples or discussion around how it is applies in our daily routines or roles. I think then that would be easier to see its application*
(LH07 postSETS)

However, one HCA explained how on one occasion since the training, she had spoken to a locum doctor. She was unsure if she used SBAR but recalls being ‘*direct*’ and approaching the situation with confidence, which she said was possibly helped by the training:

*I think I felt good, I felt confident. I do remember saying to myself, be straight forward, think about what they and the resident needs*. (LH08 postSETS)

Another HCA, despite also describing the limitations of SBAR within her role, described stepping in when a new member of nursing staff became stressed and was not managing to communicate clearly about an incident. She described drawing on the SETS training to help the colleague improve her communication:

*I said this is the way you need to do it. If you are speaking to someone and you do it this way it’s going to happen. If you just say it in like an open-ended way it’s not going to. And it is very good for teaching the differences between what an open-ended question is to what a direct communication is*. (LH04 postSETS)

None of the staff interviewed had used the SBAR paper forms, which they described as duplicating the paperwork they routinely complete after an incident, in a context where ‘*we have hundreds of paperwork*’ (LH04 postSETS):


*…accident book, and you know we have other things we need to complete in an incident like, Erm we have to complete a carers’ report, talk to the person in charge. So, I don’t think it has its place then to be honest. Too busy for it.*
(LH07 postSETS)

However, she also explained that ‘*for that information I need give over immediately* [SBAR] *is great.*’

#### 3.4.6. SBAR as a Panacea, or a Tool for Specific Jobs

When the trainer introduced the SBAR tool, he gave examples from everyday life, as well as from health and care contexts. Some interviewees agreed that it could be used ‘*every day*’ as well as when communicating about incidents or changes in residents’ states of health. However, others took the examples as literal instructions to apply SBAR in much of their workplace communication. In these cases they tended to be more critical, discussing how different approaches were needed with different people, although acknowledging that ‘*for specific things it is actually very good*’ (LH04 postSETS). They offered examples of where SBAR was not so helpful, for instance during staff handovers, where it might be necessary to repeat information, starting with an overview and then giving further details; or when showing new staff around. They further mentioned that not all carers’ English was good:


*when it is supposed to be a communication tool it can be difficult if someone doesn’t have a good grasp of English in the first place*
(LH07 postSETS)

Both of these interviewees explained that there was sometimes the need to be more ‘*personable*’ than SBAR allowed, e.g., an encouraging approach was needed when asking a resident to participate in an activity. For one interviewee, SBAR could even impede personalised, caring communication:

*Well, not sure if this is the right thing to say, but I’m going to say it, I actually find it so impersonal. I have a very chatty nature, a very personable approach and it felt a little unnatural to ask things in the way it structured. I guess it is personal preference.* […] *…it is basically a tool, but we work in such a way that really suits my caring side […] and all residents can be so different from one another*
(LH07 postSETS)

Despite these two interviewees’ reservations, they both identified situations where SBAR could be useful, which were different to those covered in the SETS scenarios. For example, one HCA had used SBAR with the family of a resident receiving palliative care; she described using it ‘*as a defensive mechanism just to keep myself at a professional level*’ in a context where she too was emotionally involved:

[The family] *…will want to know what’s going on and they will want, but it is going to hurt them and you are mentally trying to prepare yourself to give... not that we are giving them the news that this person is dying, that’s not up to us, that’s not us but during the last couple of days there would be moments where you know you’re trying to do the right thing [by providing some information] but you’re also having to protect yourself and in those cases you case use it*. (LH04 postSETS)

Her words also illustrate HCAs’ defined roles with regard to communication: they are not the ones to tell relatives that their loved one is dying.

Similarly, this interviewee, who was the most critical of SBAR, explained:


*I haven’t used it with family members, but yeah, I think it could be especially if you need to be precise, not take too long or wanted to be drawn into something because it was not good for the resident or the family. It is simple, factual and it is about sharing enough information that does not overload someone.*
(LH09 postSETS)

## 4. Discussion

### 4.1. Main Findings

We found that two very different simulation-based training sessions were acceptable to HCAs working in nursing homes, and could be delivered in the workplace.

HCAs described benefits to both types of training, including potential improvements in practice. They expressed dedication to their roles, and assured researchers that they were already well able to care for residents with dementia and to communicate effectively in the workplace (and researchers had no reason to doubt this). This contrasted with HCAs’ considerable nervousness and under-confidence about demonstrating such skills in front of colleagues in the SETS role-plays. Despite this, HCAs found the SETS feedback discussions interesting and valuable, and specifically mentioned how debriefings after each role-play scenario aided learning, and the trainer’s manner put them at ease. In contrast, AWTD—a self-contained digital package—required no interpretation when used by dementia-trained HCAs. HCAs felt that AWTD did not provide new knowledge (beyond what they already knew), but brought the experience of living with dementia to life, and so may enhance person-centred care.

AWTD was quick and easy to deploy in care home settings, requiring minimal additional resources. SETS was more resource-intensive and time-consuming, and therefore cannot be implemented ad hoc, which presents challenges to uptake in care homes, where staff changes make it difficult to anticipate who can attend on a particular day. HCAs’ perceptions of the value of SETS was somewhat limited by their views SBAR’s utility. First, there was a strong sense that communicating with external professionals was outside of the HCA role. Such tightly defined roles may prevent HCAs from developing their clinical leadership abilities, with impacts on residents’ care. Second, they described needing to provide information in different ways when communicating with colleagues and residents, to repeat information, and to be personable—related to their caring role and comprehension difficulties. Self-consciousness about their own English may have contribute to nervousness about SETS role-play participation.

### 4.2. Discussion of Findings in the Context of Existing Research

Simulation-based training evolved in hazardous professions such as aviation, to maximise training safety and minimise risk, and has only relatively recently been used in nursing practice. Our study contributes to the limited research on the use and impact of simulation training in care home settings. Of three recent studies, one focuses on ethical dilemmas in caring for persons living with dementia, and suggested that simulation training helps nursing students to adapt to these situations in clinical practice. The exposure that simulation gives in a supportive learning environment helps to foster security in learning, but as we have shown, the facilitator’s role is important in the creation of this environment [54]. The second study (linked with ours, concerning SETS) explored the feasibility of delivering in situ simulation *within* care homes, concluding that simulation is acceptable to staff and leads to increased knowledge on the recognition and management of common conditions in older people [55]. The final study was small and highly specific in its use of simulation: concerning care home staff’s ability to facilitate advance care planning for patients with advanced dementia [56].

Supporting our findings about the discussions within SETS training, others have identified the importance of the human facilitator’s role in debriefing role-play simulations, which is key to the success of such training [57]. The need for skilled facilitation, and the barrier to implementation posed by limited staff time, are not unique to SETS or to simulation-based training; they have been found for diverse types of training in care home settings [58]. We found that SETS was feasible to implement with HCAs, with the support of the care organisation which runs the homes and home managers. Indeed, SETS has been delivered in a large number of care homes, with participants valuing the discussion elements within the debriefing [55]. SETS is currently funded and so its implementation required no financial outlay from the homes, however we acknowledge that if this situation were to change, care home organisations might have to find funding for such a scheme, with implications for feasibility.

AWTD has been used in education programmes in the UK and elsewhere. An evaluation of AWTD and an accompanying workbook (the latter unavailable at the time of our study) has been undertaken [59], but is not yet published, and we have found no other published evaluations. Other interventions using VR to simulate experiences of dementia have been studied, with similar findings to our own, particularly in terms of increased empathy and understanding of what it is like to live with dementia. Slater et al. evaluated the Virtual Dementia Tour^®^ (VDT), a ‘sensory distortion programme’ where vision, touch and sound are distorted through use of goggles, shoe inserts, gloves and headphones, and facilitators subject participants to curtness and aloofness to simulate the experiences of people with dementia in healthcare environments, followed by a debriefing session [60]. Evaluation participants included 72 health care personnel and community and family carers, in hospitals, community and voluntary services, in the Republic of Ireland. VDT was found to enhance a sense of empathy among participants, which led to greater confidence, compassion and person-centred practice. The Dutch ‘Through the D’mentia Lens’ (TDL), a simulation movie played on a VR device, accompanied by an online course, has been evaluated through surveys with informal/family carers, in pilot study with a before-and-after design [61]. After experiencing TDL, informal carers were more empathic, and felt that they understood better what it was like to have dementia and the perceptions of people with dementia. In contrast to our findings and those of Slater et al., they found no change in person-centredness, perhaps because informal/family caring relationships may already be highly person-centred. A multimedia arts exhibit in Canada, about the experience of living with dementia, including VR, was evaluated using mixed methods. Researchers concluded that it increased empathy and understanding of dementia among nursing students, with VR being among the most engaging media [62].

SBAR itself may usefully lead to a common language between healthcare providers, increasing confidence in communication and ultimately leading to more efficient communication [63], as our findings suggest. Studies of SBAR in care home settings demonstrate that staff consider it potentially useful, and that it may provide cues for effective communication [64], but key champions are needed to ensure its success as a tool at handover more generally [65].

### 4.3. Strengths and Limitations

A strength of our qualitative evaluation is that we trialed two contrasting forms of simulation-based training, enabling us to explore the benefits common to both types of training, and thus draw out tentative findings about the use of simulation per se with HCAs in nursing home settings. In addition to the contrasts noted in the introduction (Section 1.4), we found that whilst one training topic was familiar (dementia), the other (SBAR) was unfamiliar to HCAs. Our use of two nursing care homes run by the same organisation in the same region, both providing specialist dementia care, may limit the transferability of our findings to other older people’s care settings. Differences with other settings may include availability of existing training (existence of the Academy perhaps indicates greater investment in staff training than smaller care home providers can offer). However, we have no reason to believe that the care homes were especially unusual.

The position of HCAs in nursing homes and care homes globally is hugely diverse; across European countries they are referred to by up to 18 different titles, and their education and training is also enormously varied [66]. The transferability of our findings to HCAs in nursing homes outside the UK is therefore difficult to assess; we could make no comparisons or contextualise this study to HCA experiences in other countries. 

We experienced no challenges in engaging staff with AWTD training, which was quick and easy to deliver. However, although we and the care home managers made efforts to enable and encourage SETS attendance, neither course was full, and despite multiple visits to both homes we experienced challenges in obtaining post-training interviews. These issues relate to staffing challenges which are typical of the care home sector: staff were busy and unavailable at short notice (e.g., needed to provide cover, or had changed shifts), and some had moved on. Whilst we were able to obtain valuable feedback on the experience and impact of both types of training, and to identify themes across the post-training interview dataset—encompassing shared and divergent views and experiences—we may not have achieved thematic saturation. In future research with staff in these settings we could explore measures to increase post-training interview participation, such as offering interviews outside of staff’s work time, perhaps by telephone or online and/or with an incentive/voucher.

### 4.4. Future Directions for Practice and Research

We suggest that due to the minimal resources and staff time required, and its ease of use, AWTD could be used in the induction of HCAs new to working with people with dementia, and in conjunction with existing dementia training, it may enhance the ability of HCAs with and without prior experience of dementia care to provide person-centred care.

Our study was conducted before the COVID-19 pandemic. The impact of COVID-19 on older care homes residents [67] is well-documented, whilst COVID-19-related morbidity and mortality have been high amongst people living with dementia, who are also at increased risk of neuropsychiatric disturbances due lockdown and the social isolation measures which have been applied stringently in nursing homes [68,69,70]. During the pandemic, nursing home staff have been at considerable personal risk [67], exacerbating the challenges already experienced in this sector (see Introduction). Enhanced training and support for new and existing staff are needed, to grow and develop the nursing home workforce [71]. Simulation-based training, which does not involve contact with residents, may play an important role in the pandemic context, and this requires further exploration. The pandemic has also led to changes in ways of working across older people’s care—in the community as well as in care home settings—and people who do not usually work as carers (e.g., cleaners, council workers) have sometimes taken on caring roles [72]. As these people may lack awareness of dementia and its effects, AWTD may be helpful in this context.

Further research could explore the barriers and facilitators to giving HCAs a greater role in clinical communication, supported by training such as SETS.

As existing studies on simulation for dementia awareness and clinical communication training with care staff have limited generalisability and/or lack long-term objective outcome measures, larger-scale mixed-methods evaluation of the two trainings is warranted. These could be undertaken with a more diverse range of care settings, trainees and resident/patient populations.

## 5. Conclusions

Simulation-based training, delivered in the workplace, is an acceptable and impactful means of skills development training for HCAs working in nursing homes. Our findings suggest that in these settings, AWTD may be effective in enhancing person-centred care as a complement to existing dementia training, requiring little staff time or resources to implement. SETS may improve communication with colleagues and other professionals, however this training requires more resources, and time, co-ordination and commitment from both managers and care staff to attend the training.

## Figures and Tables

**Figure 1 ijerph-18-03995-f001:**
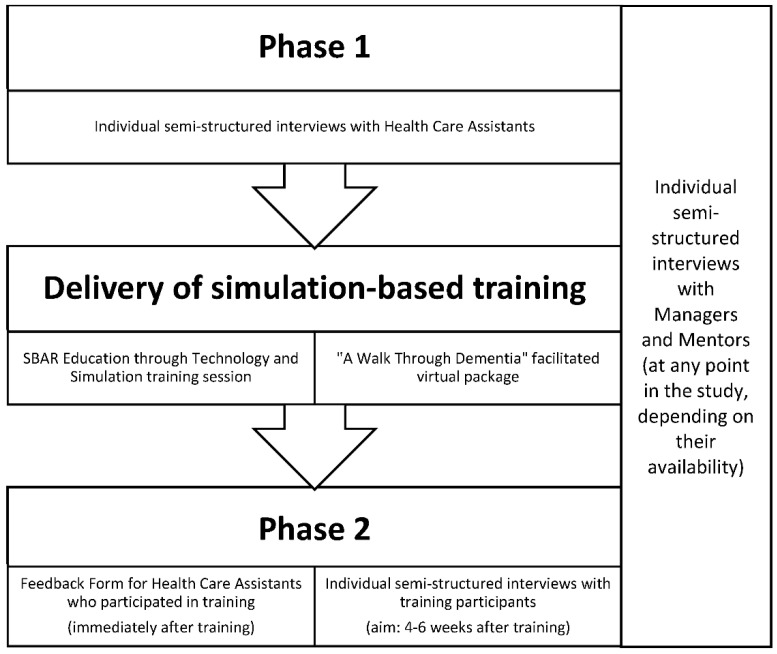
Flowchart showing study data collection in relation to training activities.

**Table 1 ijerph-18-03995-t001:** Interview participants’ current job role, phase of study participation and training undertaken.

	HCA	Nurse	Manager or Assistant Manager *	Total Interviews
Phase 1: pre-training interviews (including manager/mentor interviews)	10	1	4	15
Phase 2: post-training interviews	7 **	1	1	9
*Training undertaken by phase 2 interviewees:* *AWTD* *SETS*	*2* *5*	*1* *0*	*1* *0*	*4* *5*
Total interviews	17	2	5	24

* One Assistant manager was concurrently working in a nursing role at the home (as Clinical Lead); another Manager had a nursing background. ** Including one currently working as an Activities Co-ordinator in the home.

## Data Availability

The data presented in this study are available on reasonable request from the corresponding author. The data are not publicly available due to ethical reasons concerning participant confidentiality (although reasonable steps have been taken to de-identify the data, it is possible that research participants might be identifiable to their colleagues).

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
