# Peer review of "‘This Adds Another Perspective’: Qualitative Descriptive Study Evaluating Simulation-Based Training for Health Care Assistants, to Enhance the Quality of Care in Nursing Homes"

_ijerph, 2021, doi:10.3390/ijerph18083995_

Round 1

Reviewer 1 Report

The authors have improved the manuscript and included new data.

Author Response

 Thank you for your comments

Reviewer 2 Report

  • Suggest providing more context about National Health Service, as many readers will not be familiar with UK system.
  • Grammatical errors throughout, please proofread.
  • No need to introduce HCA abbreviation twice.
  • clarify the connection between educational opportunities and person-centered care.
  • articulate the connection between person-centered care and SBAR.
  • suggest adding in description of AWTD prior to aims like authors did with SBAR.
  • nice job adding in rationale for study design
  • recommend including a 1-2 sentences at beginning of methods that describes the study as a 2 phase study.
  • Provide rationale for why interviews with managers and mentors needed to occur throughout the study.
  • Recommend describing study as mixed methods if quantitative data were gathered; or remove quantitative data to avoid confusion....in particular, how can authors claim that quantitative data analysis was not undertaken, but then in previous sentence, authors state that quantitative data from feedback forms corroborated qualitative interview findings? Need to revisit methods and explore mixed methods research.
  • Table 1 is a nice addition that clarifies sample
  • Suggest using either SBAR or SETS to limit use of abbreviations since the SBAR was formatted as SETS.
  • aims of study do not clearly get at feasibility of delivery of simulation-based training, rather acceptability. Yet, results and discussion mention aspects of feasibility. There is still a disconnect in language used across sections of the manuscript.
  • Appreciate connection to COVID; suggest elaborating a bit more on how pandemic has influenced care delivery.

Author Response

Please see in the attachment

This manuscript is a resubmission of an earlier submission. The following is a list of the peer review reports and author responses from that submission.

Round 1

Reviewer 1 Report

The need of well trained professionals in nursing care home settings is an urgent need, as from some years there has been an continuous increase of this population and they care is far from optimal.

Therefore, any initiative directed to improve the preparedness of the different professionals involved in elderly care is welcome.

I found very interesting the use of simulations in this training, as it is a practical approach that usually is not used.

I consider that this manuscript represents an important input in this area. However, I think that the authors should include more information of some aspects:

  • What is the professional background or formation of the HCA? Is homogenous or there are different profiles?
  • Methods:
    • I found that there should be included some information about who are the people doing this formation and what experience do they have in this field.
    • What were the principal characteristics of the institutions where the simulations were done? Number of inmates, some of their characteristics.
  • Results:
    • How many HCA attended the different simulations?

Reviewer 2 Report

Summary:

This manuscript discusses an interesting topic that has limited evidence—perceptions on use of dementia care simulation trainings in nursing home settings. I appreciate the authors’ attempt to explore the acceptability of two different types of trainings among health care assistants and other nursing care staff/leadership. However, there are many areas for improvement throughout the manuscript. While the authors do a nice job describing the need for simulation-based trainings in the health care sector, there is a lack of a clear, concise description of the two selected trainings (i.e., AWTD and SETS). The methods section also lacked necessary details, such as selected design and rationale, sampling strategy, identification and training of person responsible for conducting interviews, inclusion criteria for participants, use of thematic analysis process, and incorporation of strategies to increase rigor of study. The authors may need to revisit the data, dive a bit deeper into their interpretation of the results, and consider the suggestions below to help improve clarity and strengthen  the  overall  message  of  the  manuscript.

Abstract:

  • Specify qualitative research design (e.g., phenomenology, basic qualitative description)
  • Indicate any methods/strategies used to enhance trustworthiness of data/findings
  • Suggest limiting use of abbreviations to increase readability

Introduction:

  • Consider starting the introduction with dementia focus, presenting descriptive statistics, rather than the ageing focus
    • Also, consider how study could be better described to align with scope of journal (ie., quality of our environment, the interrelationships between environmental health and the quality of life, environmental medicine, and public health)
  • Suggest only presenting information on nursing homes, not residential care homes, to avoid confusion between the two types of settings
  • Specify why nursing home care falls outside of the country’s National Health Service
  • Suggest using the term ‘residents’ throughout manuscript, rather than ‘older care home residents’ and ‘nursing homes’ rather than ‘nursing care homes’
  • Check for grammatical errors
  • Provide background information on why there is a gap in education and training of HCA (e.g., why doe new starters often lack appropriate leadership skills and have to learn on the job?)
    • Consider combining ideas in paragraphs 3 and 4 to help address previous comment; what degrees and/or previous professional experiences are required in the UK to obtain a HCA position?
  • Say more about how the research included in the systematic review of SBARs is lacking ‘quality’, if moderate evidence was found
  • Improve connection/transition sentence between SBAR and simulation paragraphs
  • Specify type of review for reference 35
  • Provide potential explanation for why there is limited evidence for use of simulation training in nursing home care settings (e.g., does this have to do with the shortage of funding mentioned in paragraph 4?)

Materials and Methods:

  • Indicate type of qualitative design and rationale for selection of design (include references); it is not sufficient to describe study as a ‘qualitative evaluation design’
  • Use UK, or England throughout
  • Provide sampling strategy (e.g., purposive?)
  • Specify if authors received funding to provide simulation training to nursing home staff for free
  • Describe nursing home care settings more (e.g., resident/staff ratio, location (urban, rural), years in service). How do the two settings compare?
  • Describe importance of staff using the term ‘mentor’, especially if that was a term assigned by research team. Regardless of title, it appears that the nursing home settings have assistant managers, clinical lead nurses, and some HCA supervisors that take on training of front-line care staff
  • Share that questionnaire was used in Phase 2 to obtain feedback on training; suggest describing this questionnaire in data collection section in addition to the semi-structured interviews
  • Describe how interview guides and questionnaire were developed (e.g., theoretical framework focused on training, or existing literature?)
  • Explain decision to interview/train participants during regular working hours. Could this not have been done on a weekend with a stipend provided to the participants to increase participation rate?
  • Review basic qualitative description approach (Sandelowski, 2000) and determine if data collection/analysis aligns with that design; if feedback forms were used to corroborate qualitative findings, this study aligns with the convergent parallel mixed methods design (Creswell & Plano-Clark, 2011)
  • Provide reference for AWTD app
  • Clarify sentence “In every scenario there was a need to communicate with another person (e.g. GP, resident’s relative) and the trainer took this role.”
  • Describe the Team GAINS debriefing model. Why was that model used? Is it part of the SETS training? What does GAINS stand for?
  • Specify whether the research team used any data management software and transcription services
  • Include rationale for selection of thematic analysis
  • Share whether or not the researchers that completed analysis had any previous qualitative experience or training
  • Provide any information on ways to increase trustworthiness of findings (e.g., peer debriefings, audit trail, member checking, negative case analysis)

Results:

  • Consider creating a Table 1 to illustrate demographics of study participants (e.g., gender, race/ethnicity, years in position, title), and involvement in training/data collection. Then you can specify N and n within table columns. It is difficult to follow within the text
  • Provide more direct quotes from study participants to emphasize subjective experience/insight, rather than researchers’ interpretation in sections 3.1 and 3.2
  • Clarify link between introduction and results section (e.g. in introduction, authors specify that there is a gap in training for HCAs in nursing home settings; however, in results, study participants described quite a few training opportunities available to them through their employers)
  • Clarify whether the consent form for study participation included information on role play and potential physical reactions to simulation (e.g. wouldn’t these study characteristics need to be communicated to participants ahead of time?)
  • Consider providing background/professional information on actor that was part of simulation training in the methods section
  • Consider combining sections 3.4 and 3.3—there are many similarities within these sections
  • Consider breaking this manuscript into 2 to clearly differentiate need for, and findings from SBAR training from simulation training; authors have access to so much rich data that they could use to hone in on some different areas
  • Check for grammatical errors
  • Given the quotes that were shared in the results section on SBAR, it would be helpful to have learned more about the tool in the methods section, including the steps and communication examples

Discussion:

  • Given the opening summary statement in the discussion, should the study aims be focused on evaluating the acceptability and feasibility of the AWTD and SETS in nursing home care?
    • Specify how feasibility was assessed. Based on methods, the researchers delivered the trainings, not the settings/staff. Limited information presented on how the settings/staff could implement the trainings in the future. There appears to be a disconnect between presented results and elaboration of ideas in discussion.
    • Research team implemented the trainings, so can the authors extrapolate that the trainings would be feasible?
  • Suggest connecting need for person-centered care to introduction and results section to better align with key point in discussion.
  • Given low response rate mentioned in limitations section, can authors conclude that trainings are feasible and “quick and easy to deploy in care home settings”?

References:

  • Suggest including references from targeted submission journal (e.g. International Journal of Environmental Research and Public Health)

Reviewer 3 Report

Title: It would be convenient to review it. That shows the main idea of ​​the study and identifies the qualitative method carried out.

Abstract: It must show the relevant aspects of the study and its results.
Keywords: Nursing Care homes is not a Mesh descriptor.

Introduction: It shows a background of the subject only focused on the United Kingdom without describing the problem at a global level, with general figures that bring the reader closer to the true dimension of the problem to be addressed.
It does not identify the type of training of health assistants (HCA) and their skills. Something fundamental to be able to differentiate them from the competences of nurses.
It would need to show the kind of UK regulation that enables these workers to practice over patients in nursing homes. It is difficult to understand that a professional who is dedicated to the home care of the elderly is not certified, much less that they have literacy or numeracy problems.
You do not correctly enter the justification for your study. It raises review studies in a disjointed way in the text that do not respond to the training needs of home care personnel in the simulation learning environment.
The main objective of your study is not clear and concise. Difficult to measure. In no case can you assume that the training of HCAs will modify the role or competencies of these professionals after training. And less when those results are not measured objectively.

Material and methods:
The qualitative design of the study must be concise. It is not enough to indicate that it is qualitative.
In recruitment mix relevant information for the study. It recruits HCA staff and nurses that it identifies as mentors when the study objective is focused on HCA training only.
Nurses have sufficient training, education and competencies to carry out care in home settings without the need for any additional training.
It is not understood why interviews are also conducted with managers and mentors.
Training carried out with HCAs is not shown. Contents, duration, learning objectives, who teaches it? How? ...
It is not clear how many HCAs were trained or whether they were the same in the pre- and post-training interviews.
The data analysis is incomplete and poor. They collected quantitative data but they do not show it because of a small size, what small size is it? What was the degree of completion of the interviews before and after the training? How many HCAs completed the training? Were they always the same?
Ethical considerations: The corresponding Ethics Committee must show the study identification code.

Results.
Expressing the findings by sections or topics can be of great help. It should be supported by tables or figures to facilitate the extraction of the results of the interviews.
Phase 1 shows that 15 interviews were carried out, of which only 10 were at the HCAs. It would be necessary to know how many HCAs and nurses the residence had, as this number of interviews carried out is very low.
It is especially difficult to attribute an impact to the training carried out with only 10 interviews with the staff of a healthcare center. These findings should be analyzed quantitatively with a representative sample representation of the HCA of the healthcare center.

Discussion.
He discusses the importance and benefits of simulation-based training in settings similar to his study, but does not compare the results obtained with other similar studies.
Discuss the need for specific training in insane patients. This concept had not been introduced before in his manuscript. The training of caregivers in insane patients must be specific and very oriented to this problem; it cannot be a generic approach to training in clinical simulation care.

Conclusions
Incomplete with respect to the objectives set out in their study.